# PDE9A Promotes Calcium-Handling Dysfunction in Right Heart Failure via cGMP–PKG Pathway Suppression: A Mechanistic and Therapeutic Review

**DOI:** 10.3390/ijms26136361

**Published:** 2025-07-01

**Authors:** Spencer Thatcher, Arbab Khalid, Abu-Bakr Ahmed, Randeep Gill, Ali Kia

**Affiliations:** 1Kirk Kerkorian School of Medicine at UNLV, Las Vegas, NV 89106, USA; thatcs1@unlv.nevad.edu (S.T.); ahmeda15@unlv.nevada.edu (A.-B.A.); 2Department of Internal Medicine, Kirk Kerkorian School of Medicine at UNLV, Las Vegas, NV 89106, USA; arbab.khalid@unlv.edu (A.K.); randeep.gill@unlv.edu (R.G.)

**Keywords:** phosphodiesterase 9A (PDE9A), cyclic guanosine monophosphate–protein kinase G signaling (cGMP–PKG signaling), right ventricular failure (RHF), calcium-handling dysfunction, sarcoplasmic reticulum (SR) calcium reuptake, myocardial fibrosis, cardiac hypertrophy, natriuretic peptide (NP) signaling

## Abstract

Right heart failure (RHF) is a major cause of morbidity and mortality, often resulting from pulmonary arterial hypertension and characterized by impaired calcium (Ca^2+^) handling and maladaptive remodeling. Phosphodiesterase 9A (PDE9A), a cGMP-specific phosphodiesterase, has been proposed as a potential contributor to RHF pathogenesis by suppressing the cardioprotective cGMP–PKG signaling pathway—a conclusion largely extrapolated from left-sided heart failure models. This review examines existing evidence regarding PDE9A’s role in RHF, focusing on its effects on intracellular calcium cycling, fibrosis, hypertrophy, and contractile dysfunction. Data from preclinical models demonstrate that pathological stress upregulates PDE9A expression in cardiomyocytes, leading to diminished PKG activation, impaired SERCA2a function, RyR2 instability, and increased arrhythmogenic Ca^2+^ leak. Pharmacological or genetic inhibition of PDE9A restores cGMP signaling, improves calcium handling, attenuates hypertrophic and fibrotic remodeling, and enhances ventricular compliance. Early-phase clinical studies in heart failure populations suggest that PDE9A inhibitors are well tolerated and effectively augment cGMP levels, although dedicated trials in RHF are still needed. Overall, these findings indicate that targeting PDE9A may represent a promising therapeutic strategy to improve outcomes in RHF by directly addressing the molecular mechanisms underlying calcium mishandling and myocardial remodeling.

## 1. Introduction

Right ventricular (RV) failure is a critical determinant of prognosis in pulmonary arterial hypertension (PAH) and other diseases imposing pressure overload on the right heart [1]. Unlike the left ventricle, the thinner-walled RV is adapted for a low-pressure pulmonary circulation and often responds poorly to chronic pressure stress. Initially, the RV undergoes adaptive hypertrophy to preserve output, but prolonged pressure overload can lead to maladaptive remodeling characterized by chamber dilation, fibrosis, and progression into RHF [1]. Clinically, RHF manifests as systemic venous congestion and reduced cardiac output and is associated with high morbidity and mortality.

On a cellular level, the failing RV myocardium exhibits hallmark disturbances in excitation–contraction coupling and Ca^2+^ homeostasis, similar to left-sided heart failure. These include reduced sarcoplasmic reticulum (SR) Ca^2+^ reuptake due to downregulation of SR Ca^2+^-ATPase (SERCA2a), diminished phospholamban (PLB) phosphorylation, further impairing SERCA2a Ca^2+^ re-uptake, and enhanced diastolic Ca^2+^ leak from dysfunctional ryanodine receptor (RyR2) channels [2]. The net effect is elevated diastolic cytosolic Ca^2+^ and depleted SR Ca^2+^ stores, impairing both systolic contraction and relaxation and promoting arrhythmias [3]. Maladaptive Ca^2+^ handling also activates pro-hypertrophic and pro-fibrotic signaling pathways, such as the calcineurin–NFAT and Ca^2+^/calmodulin-dependent protein kinase II (CaMKII) axes, driving structural remodeling of the RV.

A growing body of evidence suggests that disrupted cyclic nucleotide signaling—specifically, a deficit in the cardioprotective cGMP–PKG pathway—contributes to Ca^2+^ mishandling in RHF. Cyclic GMP, generated by nitric oxide (NO)-stimulated soluble guanylate cyclase and natriuretic peptide (NP)-stimulated particulate guanylate cyclase, activates PKG, which phosphorylates multiple targets in cardiomyocytes to reduce intracellular Ca^2+^ load and blunt hypertrophic responses [4]. PKG-mediated phosphorylation of troponin I and titin improves diastolic relaxation, while PKG also modulates Ca^2+^ cycling by phosphorylating PLB, placing it in an active state to cease inhibitory functions against SERCA2a activity and by stabilizing RyR2 channels [3].

However, in heart failure, NO bioavailability is often reduced, and the NP–cGMP pathway becomes a critical alternative route for PKG activation. The partial efficacy of phosphodiesterase type 5 inhibitors (PDE5i) such as sildenafil in PH-associated RHF—by boosting cGMP—highlights the importance of maintaining cGMP signaling for RV function, yet NO-dependent cGMP generation is often blunted in failing hearts, limiting PDE5i effectiveness [5]. This has prompted interest in augmenting NP-driven cGMP signals.

Phosphodiesterase 9A (PDE9A) was first identified in the late 1990s as a distinct, high-affinity cGMP-specific phosphodiesterase [6]. Kinetic analyses showed PDE9A has one of the highest known affinities for cGMP (Km ~170 nM, with ~1000-fold lower affinity for cAMP) [6,7]. PDE9A mRNA is widely expressed across tissues (originally detected in all tested organs, with notable levels in brain, intestine, and spleen) [6], and it lacks the allosteric cGMP-binding regulatory domains found in other cGMP PDEs (e.g., the GAF domains of PDE2/5) [6]. Early studies in the brain revealed that PDE9A is the only cGMP-specific PDE substantially expressed in the forebrain, where it colocalizes with NO–soluble guanylyl cyclase signaling pathways [8]. This suggested an important role in modulating intracellular cGMP linked to neurotransmission and memory, aligning with initial investigations of PDE9 inhibitors for cognitive enhancement [8]. Cardiovascular interest in PDE9A arose when it became clear that PDE9A is present in the heart and can modulate cardiac cGMP pools distinct from those regulated by PDE5A. Notably, PDE9A expression is low at baseline in the myocardium, but is upregulated in cardiac hypertrophy and failure [8]. PDE9A preferentially hydrolyzes cGMP generated by natriuretic peptide (NP) signaling in cardiomyocytes (rather than nitric oxide-derived cGMP), and inhibition or genetic deletion of PDE9A in preclinical models preserved cGMP–PKG signaling and blunted pathological remodeling under stress [8]. By the mid-2010s, these landmark findings—from the enzyme’s molecular characterization to its unique regulation of the NO-independent NP–cGMP pathway in failing hearts—established PDE9A as a promising therapeutic target for cardiovascular diseases.

Contextualizing the aforementioned factors, RHF represents a convergence of two pathological processes: defective Ca^2+^ handling and impaired cGMP–PKG signaling. PDE9A lies at this intersection by virtue of its cGMP hydrolase activity, linking neurohormonal stress to Ca^2+^ dysregulation. The sections that follow review the following: (1) the evidence for PDE9A expression changes in the RV during stress and failure; (2) the impact of PDE9A on cardiac cGMP–PKG signaling and downstream Ca^2+^ handling; (3) alterations in key Ca^2+^-handling proteins (SERCA2a, PLB, RyR2) observed in RHF; (4) how these molecular changes translate into fibrosis, hypertrophy, and contractile dysfunction; and (5) the therapeutic implications of targeting PDE9A in RHF, including recent preclinical and clinical advancements. Importantly, nearly all published evidence for PDE9A’s role in heart disease is derived from left-ventricular models [8,9]. As such, any extrapolation to right-ventricular pathology remains hypothetical and should be interpreted with caution until supported by direct RV-specific data.

## 2. Discussion

### 2.1. Foundational Characterization of PDE9A and Its Early Roles in Disease

The discovery of PDE9A’s role in the heart is relatively recent. Under physiological conditions, cardiac PDE9A expression is low to undetectable, with minimal activity in healthy ventricles. However, under pathological stress, PDE9A levels can rise dramatically. Studies of left ventricular (LV) disease first revealed this phenomenon: failing human hearts and animal models of hypertrophy show robust upregulation of PDE9A mRNA and protein compared to non-failing controls [8,10,11]. Specifically, pressure overload via transverse aortic constriction (TAC) or hypertrophic stimulation with phenylephrine markedly increases PDE9A expression in murine hearts [5,10]. In human specimens, hearts from patients with dilated cardiomyopathy or aortic stenosis also exhibit significantly higher PDE9A protein expression and enzymatic activity compared to non-failing hearts [10,11,12]. Notably, PDE9A levels are especially elevated in hearts from patients with heart failure with preserved ejection fraction (HFpEF), suggesting a strong association between natriuretic peptide-driven signaling and PDE9A upregulation [13].

Phosphodiesterase 9A (PDE9A) was first identified in the late 1990s as a novel member of the PDE enzyme family. Fisher et al. (1998) cloned a full-length human PDE9A cDNA (593 amino acids) and noted that its catalytic domain sequence was only ~28–34% identical to those of known PDEs, defining PDE9A as a distinct family [6]. Northern blot analysis showed PDE9A mRNA (~2.0 kb) is broadly expressed across tissues, with highest levels in spleen, small intestine, and brain [6]. Biochemically, recombinant PDE9A was found to be a high-affinity, cGMP-specific phosphodiesterase—it hydrolyzes cyclic GMP with a K_m ≈170 nM (versus ≈230 μM for cAMP), making it one of the highest-affinity PDEs known [6]. Notably, PDE9A lacked the regulatory allosteric cGMP-binding domains present in other cGMP-specific PDEs (like PDE2, PDE5, PDE6) and was insensitive to broad-spectrum PDE inhibitors such as IBMX, rolipram or vinpocetine (up to 100 μM) [6]. It was weakly inhibited by the cGMP PDE inhibitor zaprinast (IC_50 ~35 μM) [6]. These early findings established PDE9A as a unique, cGMP-selective enzyme with unusually high substrate affinity and distinctive regulatory features [6].

#### 2.1.1. Molecular Structure and Substrate Specificity

Structurally, PDE9A’s catalytic domain shares only ~28% amino acid identity with PDE5A [6], despite both enzymes being cGMP-specific. The first crystal structure of the PDE9A catalytic domain (human PDE9A2) was solved by Huai et al. (2004), complexed with the general PDE inhibitor IBMX [14]. This PDE9A structure revealed an active site configuration accommodating IBMX in multiple orientations [14], providing insight into the enzyme’s inhibitor interactions and high cGMP affinity. Consistent with its enzymology, PDE9A’s active site is optimized for cGMP recognition, explaining why among all cGMP-hydrolyzing PDEs, PDE9A exhibits the highest affinity for cGMP (K_m in the low nanomolar range) with essentially no cross-reactivity toward cAMP [6]. Furthermore, the enzyme requires divalent metal cofactors for activity and shows maximal catalytic rates in the presence of Mn^2+^ over Mg^2+^ [6]. Overall, these structural and biochemical characteristics underscored PDE9A’s role as a dedicated cGMP hydrolase distinct from other PDE families.

#### 2.1.2. Gene Structure and Isoforms of PDE9A

Early molecular biology studies uncovered the fact that the human PDE9A gene undergoes extensive alternative splicing, producing multiple isoforms. By 2003, at least five splice variants (PDE9A1–5) had been identified [15]. Wang et al. (2003) cloned a novel variant, PDE9A5, a 492-aa protein encoded by a transcript lacking exons 2 and 5 of the 20-exon PDE9A gene [15]. Notably, PDE9A5’s enzymatic properties closely matched those of the longest isoform, PDE9A1—both are high-affinity cGMP-specific phosphodiesterases with K_m values around 0.25–0.4 μM for cGMP (and negligible cAMP hydrolysis) [15]. Small differences were observed in their maximal velocities (PDE9A5 showed a somewhat higher V_max than PDE9A1) [15], but their divalent cation dependence and inhibitor sensitivities were very similar [15]. Interestingly, different PDE9A isoforms exhibit distinct cellular localization and expression patterns. PDE9A1 (593 aa) was found to localize exclusively to the nucleus when expressed in cells, whereas the shorter PDE9A5 resides in the cytoplasm [15]. A specific pat7 nuclear localization motif in PDE9A1 directs it to the nucleus, and, indeed, endogenous PDE9A1 protein was detectable in the nucleus (but not the cytosol) of T-lymphocytes [15]. This is the first PDE family member reported to be confined to the nucleus. The expression of PDE9A1 versus other isoforms also varies by tissue: quantitative PCR revealed high PDE9A expression in immune/lymphoid organs (spleen, lymph nodes, and thymus), with PDE9A particularly enriched in T cells compared to B cells or myeloid cells [15]. The functional implication proposed was that alternative splicing of PDE9A yields isoforms targeted to different subcellular compartments, thereby conferring precise spatial control of cGMP signaling in specific cell types [15].

#### 2.1.3. Tissue Distribution and Expression Patterns

Foundational studies mapped PDE9A expression across multiple tissues, indicating a broad but variable distribution. The original cloning study reported PDE9A mRNA in all tested human tissues, with the highest levels in the spleen, gastrointestinal tract, and brain [6]. This was corroborated by later work highlighting PDE9A’s prominence in immune and lymphatic tissues (consistent with the spleen and thymus having abundant transcripts) [15]. In the brain, PDE9A is widely expressed: in situ hybridization in rodents showed strong Pde9a expression throughout the CNS, most notably in cerebellar Purkinje cells, as well as in the cortex (especially layer V), olfactory tubercle, striatum (caudate–putamen), and hippocampal pyramidal and granule neurons [16]. Double-labeling experiments confirmed PDE9A is predominantly a neuronal enzyme, with only minor expression observed in some astrocytes [16]. Interestingly, despite relatively lower transcript levels in the normal heart, PDE9A protein is present in cardiac tissue as well. (All eight tissues surveyed in 1998—which likely included heart—showed detectable PDE9A expression [6]). Thus, from early on, PDE9A was recognized as ubiquitous yet enriched in certain cell types (neurons, lymphocytes), hinting at specialized roles in those contexts.

#### 2.1.4. Roles in Neural Signaling

Given its high expression in brain regions involved in learning and memory (e.g., cortex and hippocampus), researchers hypothesized that PDE9A modulates neuronal cGMP signaling cascades. Van Staveren et al. (2002) suggested that PDE9A helps maintain low basal cGMP levels in neurons, thereby tightly regulating pathways like neurotransmitter signaling and synaptic plasticity [16]. Indeed, they found that in cerebellum, Purkinje neurons (which abundantly express PDE9A) showed almost no cGMP immunoreactivity under baseline conditions, and even a general PDE inhibitor (IBMX) failed to raise cGMP in these cells [16]. This implied that PDE9A’s IBMX-insensitive activity keeps Purkinje cell cGMP below detection unless guanylyl cyclase is strongly stimulated [16]. Such findings positioned PDE9A as a key regulator of cGMP in the brain. Early pharmacological studies reinforced this idea: PDE9A-selective inhibitors (for example, the compound BAY 73-6691) were explored for cognitive enhancement, since elevating cGMP in hippocampal circuits was found to improve memory encoding in rodent models [16,17]. While most neural studies before 2015 were preclinical, they collectively established a model in which PDE9A in neurons and glia can shape the intensity and duration of cGMP-dependent signaling (e.g., in synaptic potentiation, neuroplasticity, or neuroprotective pathways).

#### 2.1.5. Early Insights into Cardiac Signaling Roles

Although PDE9A was not initially associated with the heart, research by the mid-2010s uncovered a significant role in cardiovascular physiology. Lee et al. (2015) provided the first evidence that PDE9A is expressed in mammalian myocardium and is upregulated in disease, such as pathological cardiac hypertrophy and heart failure [10]. This study showed that cardiac PDE9A specifically modulates the pool of cGMP generated by natriuretic peptide (NP) signaling, rather than the nitric oxide (NO)–driven cGMP pool regulated by PDE5A [10]. In hypertrophied hearts, PDE9A activity was found to blunt the cGMP-mediated protective effects of endogenous NP hormones. Consequently, inhibition or genetic deletion of PDE9A enhanced cGMP signaling in cardiomyocytes and protected against stress-induced pathological remodeling (e.g., pressure overload-induced hypertrophy) [10]. Notably, PDE9A inhibition remained effective even when NO synthase was inactive, whereas PDE5 inhibitors lost efficacy under those conditions [10]. These discoveries revealed that PDE9A controls a nitric oxide-independent cGMP pathway in the heart and can drive cardiac hypertrophy when overactive [10]. The early findings thus pointed to PDE9A as a novel therapeutic target in cardiovascular disease, distinct from the well-known PDE5A, by suggesting that blocking PDE9A could augment NP/cGMP signaling and mitigate cardiac stress responses [10].

Whether a similar PDE9A upregulation occurs in the failing RV has been the subject of recent investigations. In advanced PAH, circulating B-type natriuretic peptide (BNP) levels are typically very high, which would predict increased cardiac PDE9A expression as a compensatory response to degrade NP-generated cGMP. Indirect evidence supporting this comes from gene profiling of maladaptive RV hypertrophy, where components of the cGMP–PKG pathway were differentially expressed, particularly with a modest increase in lung PDE5A expression and other changes related to cyclic nucleotide signaling, although specific PDE9A expression data were not reported [12].

Direct evidence has also begun to emerge. Kolb et al. (2021) examined a mouse model of chronic hypoxic pulmonary hypertension (a moderate RV pressure overload model) and found that, although plasma ANP was elevated, RV PDE9A expression did not significantly change, relative to normoxic controls [9]. This lack of induction may reflect the relatively mild hypertrophy and short duration of hypoxia (3 weeks) in that model, as well as potential redundancy with other cGMP-phosphodiesterases. Indeed, the same study noted only modest increases in lung PDE5A and posited that alternative PDEs might compensate in PDE9A knockout mice [12]. Although direct evidence in the RV is limited, studies of LV pressure overload have demonstrated that more severe or chronic myocardial stress leads to a time-dependent increase in PDE9A expression. Although specific measurements of RV PDE9A in human PAH are scarce, extrapolation from LV studies suggests that a failing RV (with high wall stress and NP release) would likely show increased cardiomyocyte PDE9A. Supporting this notion, experimental LV pressure overload in mice leads to a time-dependent increase in myocardial PDE9A, with both transcript and protein levels elevated over time [6].

Another important aspect is the cellular localization of PDE9A in the myocardium. In stressed hearts, PDE9A is found predominantly within cardiomyocytes, where it co-localizes with troponin T, and is largely absent in interstitial or vascular cells [6]. This contrasts with PDE5A, which is more abundant in cardiac fibroblasts and vascular tissue. The myocyte-specific expression pattern of PDE9A suggests that even modest increases in RV tissue PDE9A could have substantial effects on cardiomyocyte signaling and calcium handling.

While definitive data on PDE9A expression changes in failing RVs are still emerging, evidence from LV studies and the pathophysiology of RHF suggest that PDE9A may be similarly upregulated in the RV; however, direct experimental data remain limited, and extrapolation from left heart models should be interpreted with caution The magnitude of upregulation may depend on the severity and duration of RV overload. Mild or early-stage RV hypertrophy (as in chronic hypoxia) might not elicit a strong PDE9A response [9], whereas end-stage RHF with high BNP levels could drive substantial PDE9A induction. This scenario aligns with observations that therapies enhancing NP signaling (e.g., sacubitril/valsartan) lead to secondary increases in cGMP-degrading capacity over time as a counter-regulatory mechanism [8]. The upshot is that the failing RV likely experiences an increase in PDE9A-mediated cGMP hydrolysis within cardiomyocytes, setting the stage for compromised PKG activity and downstream Ca^2+^ dysregulation.

### 2.2. Impact of PDE9A on cGMP-PKG Signaling

PDE9A’s principal effect in cardiomyocytes is to reduce the availability of cGMP for PKG activation, particularly cGMP generated by natriuretic peptide signaling. Because PKG is a key modulator of Ca^2+^ handling and hypertrophic signaling, excessive PDE9A activity can tilt the balance towards dysfunction. As seen in Figure 1, PDE5A, predominantly hydrolyzes cGMP produced by nitric oxide (NO)-stimulated soluble guanylate cyclase (sGC) deeper in the cytosol, PDE9A however, primarily targets cGMP generated by natriuretic peptide-stimulated particulate guanylate cyclase (pGC). The reason for this is likely due to PDE9A being localized near the plasma membrane, where pGC-derived cGMP is produced, allowing it to selectively hydrolyze this pool before cGMP can diffuse into the cytosol [5,13]. This specialization means that when NO synthesis is reduced—as occurs in many forms of heart failure—PDE9A becomes the dominant regulator of cGMP in cardiomyocytes.

Experimental data indicate that inhibiting PDE9A can augment cGMP signaling, even under conditions of low NO. Lee et al. demonstrated that pharmacological PDE9A inhibition reversed cardiac stress responses in mice with pressure overload, despite concurrent NOS inhibition, whereas PDE5A inhibition alone had no benefit without NOS activity [10]. This finding established that PDE9A constrains a NO-independent (i.e., NP-driven) cGMP pool that is crucial in heart failure.

Elevated PDE9A blunts the downstream effects of natriuretic peptide (NP) signaling by degrading cGMP before it can activate PKG. Active PKG normally phosphorylates targets that alleviate cardiac stress: it desensitizes the β-adrenergic response, reduces cytosolic Ca^2+^, and inhibits pro-growth transcription pathways [1]. PKG-mediated phosphorylation attenuates β-adrenergic receptor signaling by modulating downstream effectors such as phospholamban and troponin I, thereby protecting cardiomyocytes from catecholamine-induced overstimulation. One measurable consequence of impaired PKG signaling is reduced phosphorylation of vasodilator-stimulated phosphoprotein (VASP) at Ser239, a PKG-specific site. In failing hearts with elevated PDE9A expression, VASP phosphorylation is depressed [9], indicating attenuation of PKG activity. Conversely, in experimental models, genetic deletion of PDE9A or pharmacologic inhibition restores myocardial cGMP levels and enhances PKG activation. For example, in mice subjected to transverse aortic constriction (TAC), Pde9a knockout resulted in higher cGMP levels and greater PKG-mediated VASP phosphorylation compared to wild-type controls, correlating with protection from hypertrophy and fibrosis. Similarly, acute application of a PDE9A inhibitor (PF-04449613) to isolated cardiomyocytes potentiated NP-induced cGMP accumulation and PKG activation, an effect not observed in PDE9A-deficient myocytes where basal cGMP levels were already elevated [10].

By constraining PKG, increased PDE9A activity allows Ca^2+^ cycling abnormalities and hypertrophic signaling to progress unchecked. Under normal conditions, PKG activation phosphorylates PLB at Ser16 (overlapping with the PKA site) to enhance SERCA2a-mediated Ca^2+^ uptake, and may also contribute to RyR2 stabilization [3]. In RHF, however, these protective effects are blunted: PLB remains dephosphorylated and inhibitory, slowing SR Ca^2+^ reuptake, while RyR2 channels remain hyperactive during diastole, contributing to Ca^2+^ leak and early afterdepolarizations (EADs). Elevated PDE9A pathologically promotes pro-hypertrophic signaling pathways, such as calcineurin–NFAT, by impairing PKG-mediated regulation of calcium handling.

There is evidence for crosstalk between PDE9A-regulated and PDE5A-regulated cGMP pools in the heart. In theory, simultaneous inhibition of both PDE5 and PDE9 should maximally augment cGMP signaling. A recent experimental study in mice combining a PDE5 inhibitor with a PDE9 inhibitor yielded additive improvements in cardiac function and remodeling, compared to either therapy alone [9].

In RHF management, this could translate to dual therapy addressing both arms of cGMP metabolism. Another interplay is that chronic NP elevation—such as occurs in HFrEF—may induce feedback upregulation of PDE9A over time [10]. Thus, unchecked PDE9A can act as a maladaptive compensatory response that undermines the heart’s own protective NP signaling.

The resulting dampened PKG activity leads to impaired Ca^2+^ handling—marked by slowed reuptake and increased diastolic leak—and permits hypertrophic and fibrotic signaling to progress unchecked. This mechanistic link helps explain why PDE9A inhibition has shown benefit in multiple cardiac stress models: by preserving cGMP and PKG signaling, PDE9A inhibitors help restore a more balanced and protective calcium cycling environment in the heart. Table 1 highlights these protective effects observed with PDE9A inhibition alongside the major downstream consequences of PDE9A activity on right ventricular calcium handling, hypertrophic signaling, and fibrosis.

### 2.3. Altered SERCA2a, Phospholamban, and RyR2 Activity in RHF

The downstream impact of disrupted cGMP–PKG signaling, particularly through elevated PDE9A activity, is most clearly seen in the remodeling of calcium-handling proteins within the right ventricle. Chronic RHF exhibits many of the same deleterious changes in Ca^2+^ cycling machinery observed in left-sided heart failure, ultimately leading to impaired excitation–contraction coupling and worsening cardiac function.

One key alteration is a reduction in SERCA2a-mediated reuptake of Ca^2+^ into the sarcoplasmic reticulum (SR). In failing human RV tissue from PAH patients, SERCA2a protein levels are significantly downregulated compared to donor RVs [2], and SERCA2a enzymatic activity is depressed. Compounding this problem, the endogenous inhibitor phospholamban (PLB) accumulates in its dephosphorylated, active form. Rain et al. found that PLB phosphorylation at the PKA/PKG-dependent Ser16 site was markedly reduced in PAH RV myocardium [2], leaving SERCA2a more inhibited.

The combination of lower SERCA2a expression and higher PLB activity slows cytosolic Ca^2+^ clearance during diastole, resulting in prolonged Ca^2+^ transients, impaired relaxation, and elevated diastolic [Ca^2+^]i—features that contribute to diastolic dysfunction and promote arrhythmias. Given PKG’s known role in maintaining PLB phosphorylation, the suppression of PKG activity by excessive PDE9A critically amplifies these defects.

As SR Ca^2+^ uptake declines, cardiomyocytes attempt to compensate by upregulating the sarcolemmal Na^+^–Ca^2+^ exchanger (NCX), which hastens Ca^2+^ extrusion. While detailed RV data remain limited, LV failure studies show that NCX1 expression and activity are elevated [3], and similar trends are suggested in RHF models. Although this adaptation helps preserve relaxation, it comes at a metabolic cost: Na^+^ overload and energy imbalance, which further stresses failing RV myocytes.

Perhaps the most damaging effect of impaired cGMP–PKG signaling is on the regulation of RyR2 channels. Under normal conditions, PKG contributes to RyR2 stabilization by phosphorylation at regulatory sites (e.g., Ser-2808 and Ser-2030), which reduces the likelihood of diastolic Ca^2+^ leak [22,23]. In RHF and models of pulmonary hypertension or pressure overload, RyR2 channels become hyperactive due to oxidative and phosphorylation-mediated remodeling, resulting in increased diastolic Ca^2+^ release and SR Ca^2+^ depletion [13,24,25,26].

This RyR2 dysfunction is linked to pathological post-translational modifications, notably CaMKII-mediated hyperphosphorylation at Ser2814 and oxidative stress-induced thiol modifications. A recent study by Huang et al. in a rat PAH model demonstrated that progressing RV failure was associated with enhanced RyR2 oxidation, increased CaMKII phosphorylation, and greater diastolic Ca^2+^ leak [13]. These changes were accompanied by a decline in total RyR2 protein and increased spontaneous Ca^2+^ spark frequency, correlating with the transition from compensated hypertrophy to decompensated RV failure. Therapeutic RyR2 stabilization with dantrolene reduced leak, improved contractility, and delayed RV failure progression in vivo [13].

The cumulative effect of these disruptions—impaired SERCA2a activity, reduced phospholamban phosphorylation, increased dependence on Na^+^–Ca^2+^ exchange, and diastolic RyR2 leak—is a progressive breakdown of calcium homeostasis. As SR filling declines and calcium release becomes erratic, the RV myocardium enters a vicious cycle of contractile dysfunction, impaired relaxation, and increased susceptibility to arrhythmias.

The elevated diastolic Ca^2+^ also promotes arrhythmias, a significant clinical complication of RHF. Many of these molecular disturbances—particularly involving PLB, RyR2, and calcium cycling—are plausibly downstream of impaired PKG signaling, a mechanism shown to be regulated by PDE9A in LV models. However, ECC remodeling is multifactorial, and pathways such as CaMKII activation and oxidative stress have known roles in RV dysfunction [13].

In this context, PDE9A inhibition offers a mechanistically rational strategy to restore proper Ca^2+^ handling. By enhancing PKG activation, PDE9A blockade could reverse SERCA2a dysfunction, reduce RyR2 leak, and stabilize intracellular calcium cycling—key steps in interrupting the progression of right-sided heart failure.

### 2.4. PDE9A Effects on cAMP–PKA Signaling in Cardiac Models

Phosphodiesterase-9A (PDE9A) is a cGMP-specific enzyme with negligible affinity for cAMP (~1000-fold lower), and thus does not directly hydrolyze cAMP [10]. However, cGMP modulates cAMP signaling through cGMP-activated or -inhibited PDEs, including PDE2 and PDE3, which create localized cGMP–cAMP crosstalk [27]. PDE9A, by regulating natriuretic peptide-derived cGMP, can indirectly affect these enzymes, and thereby influence PKA activity.

In failing hearts, global myocardial cAMP levels are not significantly altered by PDE9A inhibition [8], but local effects on cAMP–PKA signaling are evident. For example, PDE9A inhibition increases cGMP near the sarcoplasmic reticulum (SR), where PDE9A is localized [10], possibly altering PDE3 activity and relieving suppression of cAMP. Additionally, higher cGMP levels can activate PDE2, which reduces cAMP in certain compartments [27]. The net effect depends on the local expression balance of PDE2 and PDE3.

Importantly, PKA and PKG both phosphorylate key calcium-handling proteins like phospholamban (PLB) at Ser16. In hypertrophied neonatal cardiomyocytes, PDE9A knockdown or inhibition restored PLB-Ser16 phosphorylation after phenylephrine-induced suppression [5]. Similarly, in isoproterenol-induced heart failure models, PDE9A inhibition increased PLB phosphorylation and improved calcium reuptake via SERCA2a [5]. These effects suggest that PDE9A indirectly enhances PKA signaling under stress, even if baseline cAMP levels remain unchanged.

Taken together, while PDE9A does not directly affect cAMP hydrolysis, its inhibition can favorably modulate PKA activity via cGMP-coupled pathways, improving calcium cycling and adrenergic responsiveness in the failing myocardium.

### 2.5. Fibrosis, Hypertrophy, and Contractile Dysfunction Due to Calcium Dysregulation

The cellular Ca^2+^ disturbances outlined above serve as powerful triggers for structural remodeling of the RV. One outcome is cardiomyocyte hypertrophy—an initially compensatory growth response that becomes maladaptive. Persistent elevation of cytosolic Ca^2+^, especially diastolic Ca^2+^, activates calcineurin, a Ca^2+^/calmodulin-dependent phosphatase that dephosphorylates nuclear factor of activated T-cell (NFAT) transcription factors. Once dephosphorylated, NFAT translocates to the nucleus and drives the expression of hypertrophic genes such as fetal isoforms of contractile proteins and natriuretic peptides [28].

In RHF, the reduction in PKG activity due to high PDE9A removes an important brake on calcineurin–NFAT signaling, since PKG normally phosphorylates and inhibits components of this pathway. The result is unchecked pro-hypertrophic gene expression. CaMKII, activated by chronic Ca^2+^ elevation, provides a parallel hypertrophic signaling route by phosphorylating class II histone deacetylases (HDACs) and other transcriptional regulators, further promoting growth. Over time, the RV free wall becomes thickened (concentric hypertrophy), but also stiff and prone to ischemia, as capillary density often fails to keep up.

Another consequence of Ca^2+^ mishandling is myocardial fibrosis—the accumulation of extracellular matrix (collagen) in the RV. Several mechanisms contribute to this process. First, Ca^2+^-overloaded cardiomyocytes struggle to relax, and the prolonged mechanical tension combined with incomplete diastolic relaxation stimulates fibroblasts and activates transforming growth factor-β (TGF-β) signaling, driving collagen synthesis. Second, Ca^2+^-mediated hypertrophic signaling often includes upregulation of pro-fibrotic cytokines and growth factors.

For example, PDE9A deletion in pressure-overloaded mice was shown to suppress expression of connective tissue growth factor (CTGF) and fibronectin—key mediators of fibrosis—suggesting that excessive PDE9A activity normally promotes their induction (8). Third, the spontaneous Ca^2+^ release and resultant cardiomyocyte death (via calcium-triggered apoptosis or necrosis) leads to replacement fibrosis, as dead myocytes are replaced by scar tissue. In RHF, patchy fibrosis commonly appears in the RV free wall and interventricular septum as failure progresses. This fibrotic remodeling increases chamber stiffness, exacerbating diastolic dysfunction and elevating RV filling pressures.

Calcium-driven changes also directly impair RV contractile performance. The reduction in systolic Ca^2+^ availability—stemming from SR depletion and RyR2 leak—means each heartbeat generates less force, a hallmark of contractile dysfunction in RHF. Moreover, alterations in myofilament properties occur secondary to chronic Ca^2+^ and neurohormonal changes. In PAH RV samples, hypophosphorylation of troponin I (cTnI) and titin has been observed, which increases myofilament Ca^2+^ sensitivity and passive stiffness [2].

While greater Ca^2+^ sensitivity might initially seem beneficial, in the failing heart it paradoxically impairs relaxation and diastolic filling because the myofilaments remain activated by Ca^2+^ for longer. The increased stiffness of titin (due to reduced phosphorylation by PKA/PKG) compounds diastolic dysfunction. Together, these changes culminate in a vicious cycle: impaired contractility reduces RV output, and incomplete relaxation hampers RV filling and coronary perfusion, worsening contractile performance. Clinically, this manifests as declining exercise capacity, systemic congestion, and, eventually, cardiogenic shock.

Notably, many of these pathological processes are mitigated when normal Ca^2+^ handling is restored. In experimental models, preventing Ca^2+^ overload and leak—such as through RyR2 stabilization or increased SERCA2a activity—leads to reduced hypertrophy and fibrosis. Similarly, enhancing PKG signaling (for example, through PDE9A inhibition) confers anti-hypertrophic and anti-fibrotic effects. In one study, chronic PDE9A inhibitor therapy not only attenuated cardiac hypertrophy, but also significantly reduced interstitial fibrosis and collagen gene expression in pressure-overloaded hearts [8]. Additional evidence from multiple preclinical models supports the anti-fibrotic role of PDE9A inhibition. In the original study by Lee et al. (2015), genetic deletion of PDE9A in pressure-overloaded mice resulted in significantly less interstitial fibrosis, reduced fibronectin and CTGF expression, and improved cardiac remodeling compared to wild-type controls [10]. These findings were attributed to enhanced cGMP–PKG signaling. More recently, in an HFpEF-like model combining obesity, hypertension, and mild pressure overload, chronic PDE9A inhibition similarly reduced myocardial collagen deposition by ~60% and suppressed pro-fibrotic gene expression [28]. Additional TAC-based studies using selective PDE9A inhibitors have confirmed these anti-fibrotic effects [20]. Together, these data indicate that PDE9A inhibition mitigates fibrosis across diverse HF models by restoring PKG activity and blunting TGF-β–driven fibroblast activation.

These benefits are attributable to improved Ca^2+^ cycling and re-engagement of PKG’s anti-remodeling actions. PDE9A inhibition has also been shown to improve diastolic function—by reducing chamber stiffness—in an HFpEF-like mouse model, consistent with enhanced titin phosphorylation and improved Ca^2+^ reuptake [5]. Although extremely high doses could slightly depress systolic function (likely due to excessive cGMP in myocytes or systemic vasodilation), the overall evidence indicates that correcting Ca^2+^ handling defects can substantially rescue RV contractile performance.

### 2.6. Therapeutic Implications of PDE9A Inhibition in RHF

The recognition of PDE9A as a contributor to maladaptive remodeling has spurred investigations into PDE9A inhibition as a therapeutic strategy for heart failure. Preclinical studies provide strong proof-of-concept that targeting PDE9A can mitigate or even reverse pathological changes in the heart. In mouse models of pressure overload (e.g., TAC), both genetic deletion of the Pde9a gene and pharmacological inhibition of PDE9A have been shown to preserve cardiac function [8].

Lee et al. reported that a selective PDE9A inhibitor halted the progression of LV 358 hypertrophy and fibrosis, and even reversed pre-established dysfunction in mice under 359 chronic pressure stress [10]. Similarly, an independent group demonstrated that a novel PDE9A inhibitor (compound C33(S)) protected rats against isoproterenol-induced cardiac hypertrophy, with treated animals displaying smaller increases in heart mass and less fibrosis compared to controls [5]. These benefits were accompanied by enhanced myocardial cGMP levels and PKG activity, confirming on-target action of the drug. In these models, PDE9A inhibition largely mimicked the effects of PKG activation—reducing hypertrophic growth, improving relaxation, and attenuating fetal gene expression—consistent with the idea that it re-engages cGMP–PKG signaling.

Specific to RHF, the therapeutic potential of PDE9A inhibition remains promising but somewhat less explored. One important finding is that PDE9A suppression appears most beneficial in conditions where NP–cGMP signaling is highly activated (and, thus, PDE9A is most relevant). For instance, in models of heart failure with preserved ejection fraction (HFpEF)—which often involves pulmonary hypertension and RHF—chronic PDE9A inhibitor treatment improved diastolic function [5].

Methawasin et al. showed that long-term PDE9A inhibition in mice with combined 374 hypertensive and metabolic stress (an HFpEF-like TAC+DOCA model) lowered left 375 ventricular chamber stiffness and improved active relaxation [8]. Although extremely high doses led to a slight reduction in systolic pressure (possibly due to systemic vasodilation or over-suppression of myocyte Ca^2+^), the net effect favored improved overall hemodynamics. Translating this to RHF suggests that PDE9A inhibition could enhance RV diastolic performance by reducing stiffness from titin and fibrosis and possibly augment systolic function by restoring Ca^2+^ availability.

On the other hand, a study of chronic hypoxic PH (a milder RHF model) found that Pde9a knockout alone did not significantly improve RV metrics [9]. This outcome likely reflects the modest cGMP drive in that model—without a strong NP signal, eliminating PDE9A conferred little advantage. This underscores the fact that patient selection will be key: RHF patients with high circulating NP levels or active guanylate cyclase signaling may derive the greatest benefit from PDE9A inhibitors.

Combining PDE9A inhibition with other therapies offers a compelling approach for RHF. In PAH, standard treatment includes PDE5A inhibitors (e.g., sildenafil), to lower pulmonary vascular resistance. Because PDE5 and PDE9 act on parallel arms of cGMP metabolism—NO-driven and NP-driven, respectively—dual inhibition could maximize myocardial cGMP signaling. Preclinical evidence supports this synergy: dual PDE5/PDE9 blockade in experimental HF yielded additive improvements in cardiac function and remodeling, compared to monotherapy [9].

For RHF patients, a regimen combining sildenafil plus a PDE9A inhibitor could simultaneously unload the RV by vasodilating pulmonary arteries (via PDE5 inhibition) and strengthen the RV myocardium (via enhanced cGMP–PKG signaling through PDE9A inhibition).

Another potential combination is with neprilysin inhibition (as seen with sacubitril/valsartan therapy). By preventing NP degradation, neprilysin inhibitors raise NP levels and cGMP production, which could in turn increase the substrate for PDE9A. It has been speculated that adding a PDE9A inhibitor to neprilysin inhibition might further boost cGMP signaling and clinical outcomes [28], essentially “completing” the amplification of the NP–cGMP pathway. Clinical trials in left heart failure are beginning to explore this approach, and similar logic could apply to RHF or PH with HFpEF.

Early clinical trials of PDE9A inhibitors have reported encouraging results regarding pharmacodynamics and safety. In a Phase IIa trial (CARDINAL-HF), the PDE9A inhibitor CRD-740, was administered to patients with HFrEF, characterized by reduced left ventricular ejection fraction but without systematic assessment of right ventricular function. After four weeks, treated patients showed a significant increase in plasma and urinary cGMP compared to placebo, confirming target engagement and augmentation of NP signaling [19]. CRD-740 was well tolerated, with a safety profile comparable to placebo.

Although this trial focused on short-term biomarker endpoints, the robust rise in cGMP suggests potential clinical benefits if sustained, based on preclinical evidence. Larger trials are planned to assess effects on exercise capacity and ventricular function. Notably, the cGMP elevation with CRD-740 was observed even in patients already on neprilysin inhibitors, suggesting that PDE9A inhibition can provide incremental activation of the NP–cGMP pathway.

No clinical studies specifically in RHF or PAH patients have been published yet. However, given the overlap in pathophysiology, similar benefits are anticipated. One intriguing study in aged mice showed that PDE9A inhibition could lower pulmonary vascular resistance and improve pulmonary vasoreactivity [29]. If this translates to humans, PDE9A inhibitors could provide dual benefits in RHF by both enhancing myocardial Ca^2+^ handling and modestly reducing RV afterload.

In the context of precision cardiology, targeted PDE9A therapy opens the door to tailoring clinical treatment based on molecular profiling. An example would be RHF patients who demonstrate high BNP levels or a high NP-to-cGMP ratio (indicative of cGMP breakdown), marking them as ideal candidates for PDE9A inhibitor therapy. On the other hand, a patient whose RHF is driven mostly by low NO availability, will find PDE9A inhibition to be far less effective. Lastly, biomarkers like VASP phosphorylation or plasma cGMP levels should be considered as a potential means to help guide patient selection and therapeutic monitoring.

We explicitly acknowledge that the proposed role of PDE9A in RHF is speculative. For example, in a murine model of chronic hypoxia-induced pulmonary hypertension (a right-ventricular pressure-overload model), Pde9a deletion did not alter RV hypertrophy or cGMP signaling [9].

### 2.7. Limitations

While this review highlights the potential mechanistic role and therapeutic relevance of PDE9A in right heart failure (RHF), it is important to acknowledge key limitations in the current body of evidence. A key limitation of this review is that much of the mechanistic and therapeutic rationale for PDE9A inhibition in RHF is extrapolated from studies conducted in left ventricular (LV) or HFpEF models. While these models share important features with RHF—such as pressure overload and elevated natriuretic peptide signaling—fundamental differences in RV anatomy, loading conditions, and molecular adaptation may limit the applicability of these findings. Therefore, conclusions drawn about PDE9A’s role in RHF should be regarded as hypothesis-generating and in need of direct experimental validation. Additionally, although the right and left ventricles share common signaling pathways, particularly regarding calcium handling and cGMP–PKG regulation, they differ significantly in structure, embryology, and load sensitivity, which may affect how PDE9A functions in the RV.

Currently, direct data examining PDE9A expression or function specifically in the right ventricle under stress are sparse. For example, a study by Kolb et al. found that PDE9A deficiency did not confer benefit in a chronic hypoxia model of RHF, suggesting that the role of PDE9A may be context-dependent or less prominent in certain RV pathologies [9]. Similarly, while disruptions in RV calcium cycling have been described in PAH-related RHF [2], the specific contribution of PDE9A to these changes remains unconfirmed.

These gaps raise concerns about the generalizability of conclusions based on LV models and underscore the need for more targeted RHF research. Therefore, while PDE9A represents a promising therapeutic target, all interpretations in this review must be viewed through the lens of these current evidentiary limitations.

## 3. Materials and Methods

A literature review was performed using PubMed and Google Scholar databases. Keywords for this search included “PDE9A,” “PDE5,” “cGMP,” “GAF domain,” “nitric oxide signaling,” and “phosphodiesterase regulation.” Although the primary literature search focused on articles published from 2015 to 2025, key foundational studies published prior to 2015 were also included to ensure comprehensive coverage of PDE9A biology and its role in disease. Article types included observational studies, retrospective studies, randomized controlled trials, systematic reviews, and meta-analyses.

## 4. Future Direction

As PDE9A is emerging as a potentially important regulator of maladaptive cardiac remodeling in RHF, future studies must now define its optimal therapeutic application. Several critical gaps in knowledge must be addressed. It remains unclear which patient subgroups will benefit most from PDE9A-targeted therapy, and whether timing of intervention influences outcomes. Further investigation is also needed to clarify whether the RV is uniquely susceptible to PDE9A-driven dysfunction compared to the LV. Additionally, the regulatory mechanisms behind PDE9A upregulation—potentially influenced by inflammation, neurohormonal activation, or metabolic stress—require further study.

In order to move forward, clarification on PDE9A interventions is essential. There is a greater need for dedicated clinical trials that focus specifically on RHF and PAH populations and which examine how PDE9A inhibition interacts with standard-of-care treatments such as PDE5 inhibitors and neprilysin inhibitors, and not just simply assess structural and functional improvements. Combination regimens should be considered as a means of synergistic benefit, and deserves serious exploration. Additionally, reliable biomarkers such as NT-proBNP:cGMP ratios or myocardial VASP phosphorylation could be novel means for patient selection and therapeutic monitoring.

We must also investigate the long-term safety and durability of PDE9A inhibition in regards to chronic usage. Continued animal models, alongside increased extended human studies, will be a necessity in order to evaluate PDE9A inhibition for potential adverse effects, particularly impacts on systemic blood pressure or renal function. Additionally, PDE9A’s involvement among broader stress-signaling networks, such as CaMKII and adrenergic pathways should be investigated further. By addressing all the above mentioned, we may be able to further develop combination strategies to more comprehensively address the calcium cycling abnormalities in RHF.

## 5. Conclusions

The complexity seen in right-sided heart failure (RHF) due to disrupted calcium handling and intracellular signaling pathways has caused it to remain as a significant clinical challenge. Among the myriads of mechanical imperfections, phosphodiesterase 9A (PDE9A) has emerged as a particular target of interest for therapeutic application, as it is increasingly implicated in cardiac calcium mishandling, maladaptive hypertrophy in patients with RHF, and fibrosis. Encouraging results from both genetic deletion and pharmacologic inhibition of PDE9A suggest the possibility of substantial clinical benefit. Preclinical models have shown that PDE9A intervention shows improvement in diastolic function and reversal of adverse remodeling. With early human trials already underway, PDE9A inhibition presents a promising new approach in the treatment of RHF.

Ultimately, the therapeutic potential of PDE9A inhibition may lie in its ability to address one of the proposed molecular contributors to RHF at the molecular level. While more work remains, the current trajectory of research is encouraging. With focused clinical efforts and mechanistic insight, PDE9A inhibitors could become a valuable tool in improving outcomes for patients with this often-overlooked form of heart failure.

## Figures and Tables

**Figure 1 ijms-26-06361-f001:**
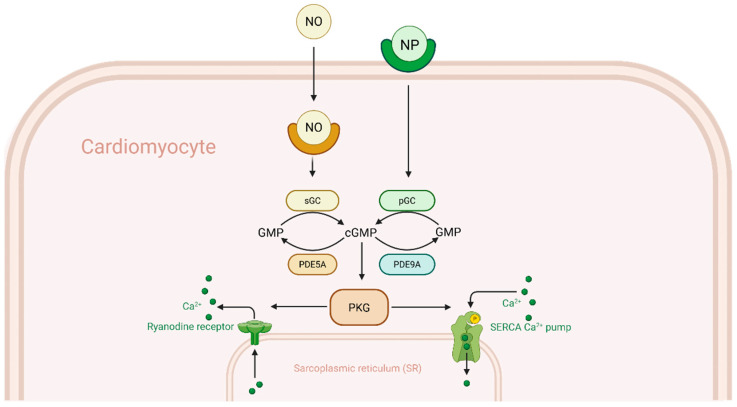
PDE9A in cGMP–PKG signaling and calcium handling in right heart failure. Natriuretic peptides (NP) and nitric oxide (NO) generate cyclic GMP (cGMP) via guanylyl cyclases (pGC and sGC), activating protein kinase G (PKG). PKG enhances calcium reuptake by phosphorylating phospholamban (PLB) and reduces RyR2-mediated calcium leak, likely by modulating its gating behavior. Phosphodiesterase 9A (PDE9A) degrades NP-derived cGMP, reducing PKG activity. In right heart failure (RHF), PDE9A upregulation impairs calcium cycling, while its inhibition restores function.

**Table 1 ijms-26-06361-t001:** Molecular Effects of PDE9A Activity in Right Heart Failure and the Benefits of PDE9A Inhibition.

Pathological Effect of PDE9A in RHF	Molecular Mechanism	Therapeutic Effect of PDE9A Inhibition
Decreased cGMP levels	Hydrolyzes NP-generated cGMP [5,10]	↑ cGMP accumulation, restored PKG signaling [10,18]
Decreased PKG activity	Loss of PLB, titin, troponin I phosphorylation [9,19]	Enhanced SERCA2a function, improved relaxation [5,18,20]
Ca^2+^ overload and leak	Impaired SERCA2a reuptake, RyR2 diastolic instability [5,21]	Stabilized Ca^2+^ cycling, reduced arrhythmias and leak [5,20]
Cardiomyocyte hypertrophy	Activation of calcineurin–NFAT and CaMKII pathways [5,10,20]	Suppressed hypertrophic gene expression [10,20]
Myocardial fibrosis	Suppression of cGMP–PKG pathway → Fibronectin, CTGF, and Collagen I/III activation [5,8,15]	Reduced collagen deposition and fibrosis via PKG-dependent antifibrotic signaling [5,8,15]

Note: Mechanisms and therapeutic effects are derived from both right ventricular (RV) and left ventricular (LV) models, including HFpEF. Most findings originate from LV models and are extrapolated to RHF; where RV-specific data are unavailable, the effects are proposed as potential consequences of PDE9A activity based on mechanistic inference.

## Data Availability

All data reported is present on PubMed, Scopus, Web of Science, and Google Scholar web databases.

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
