# Peer review of "PDE9A Promotes Calcium-Handling Dysfunction in Right Heart Failure via cGMP–PKG Pathway Suppression: A Mechanistic and Therapeutic Review"

_ijms, 2025, doi:10.3390/ijms26136361_

Round 1

Reviewer 1 Report

Comments and Suggestions for Authors

Please, see attached

Reviewer 2 Report

Comments and Suggestions for Authors

Authors: Spencer Thatcher et al.

The manuscript is a review that explores the role of Phosphodiesterase 9A (PDE9A) in the pathogenesis of right heart failure (RHF), particularly its effect on calcium handling and the cGMP–PKG pathway. It further discusses therapeutic implications and future directions. Although, the review provides a thorough discussion of molecular pathways, supported by preclinical and limited clinical evidence. Authors suggest a big role of PDE9A in the progression RHF, however there is no direct evidence of these effects in right ventricle shown in publications.

Main comments:

  1. In many times, the assertions about PDE9A in RHF are extrapolated from left heart failure models. Authors need to find any direct evidence of its role in RHF. Also a big limitation part should be written highlighting that limited direct evidence in RHF and caveats for extrapolation should be clearly stated.
  2. Limited direct evidences raises concern regarding the interpretation of the data. Authors are suggested to highlight areas of uncertainty and avoid overstating conclusions without robust data.
  1. Methods for literature search is limited to publications from 2015-2025, however, the authors should add all the information regarding the role of PDE9 in disease.
  2. Cited literature has very inconsistent numbers. The first number is 8. Please use a citation services such as Endnote, Mendeley etc. and carefully check the numbers.
  3. Table 1 lacks the links to the original research. If there isn’t these should be clearly stated as a proposed potential effects of PDE9a in RHF.
  4. The manuscript lacks summary figures that could aid in understanding complex pathways. Suggestion: Include at least one diagram illustrating the role of PDE9A in calcium handling and cGMP–PKG signaling in RHF.
Comments on the Quality of English Language
  1. There are some grammatical errors and mislabeling within the text; please proofread it.

Round 2

Reviewer 1 Report

Comments and Suggestions for Authors

While this summation of article “PDE9A Promotes Calcium Handling Dysfunction in Right Heart Failure via cGMP–PKG Pathway Suppression: A Mechanistic and Therapeutic Review” by Thatcher S. et al. is improved, it still has several issues that needs to be addressed:

  1. Reference numbering is still incorrect. For example: Line 311: Lee et al. is not ref# 7.

  1. There significant repetition on PDE9 effects on Ca handling.  Avoid such repetition would improve text flow.       

(Lines 331/332 and 351/352 are almost identical)

  1. Table 1. Please specify “pro-fibrotic genes”.

  1. Paragraph at lines 367-372 gives an impression that PDE9A is a solely decisive factor in ECC remodeling. Is this true? What about other pathways (CaMKII for example)?

  1. PKG phosphorylation ‘stabilizes RYR’. Is phosphorylation making RYR structure more stable or changes RYR sensitivity to Ca and thus reduce RYR Ca leak?

  1. Figure 1. Arrows by PDE9A and pGC point in the opposite direction, i.e. indicated that PDE9 synthesize cGMP.

Author Response

Comment 1

Reference numbering is still incorrect. For example: Line 311: Lee et al. is not ref# 7.

Response 1

Thank you for addressing this, we have been having significant technical difficulties with our Zotero. It stopped connecting to our original document and wouldn’t update when we tried to make corrections. Their official support said the only thing we could do was copy/paste to a new document and redo everything. We ended up having to correct everything by hand so we apologize for the additional errors.

We have done further proof reading and believe that all sources correlate correctly with their reference placing now.

Comment 2

There significant repetition on PDE9 effects on Ca handling.  Avoid such repetition would improve text flow.       

(Lines 331/332 and 351/352 are almost identical)

Response 2   

Thank you for your insight,

On line 350 of paragraph 4, page 8, We altered “The resulting dampened PKG activity leads to impaired Ca²⁺ handling—marked by slowed reuptake and increased diastolic leak—and permits hypertrophic and fibrotic signaling to progress unchecked.” to read as “These effects culminate in disrupted calcium homeostasis and structural remodeling, including both hypertrophy and fibrosis.” We feel this maintains continuity without rehashing the mechanism.

Comment 3

Table 1. Please specify “pro-fibrotic genes”.

Response 3

 Thank you for pointing this out. The table that begins on Page 8, Line 359 indeed was not clear enough on the specific genes we wanted to highlight from our source. We have changes Pro-fibrotic to specify Fibronectin, CTGF, and Collagen I/III  activation in order to be more clear of the genes we are referencing.

Comment 4

Paragraph at lines 367-372 gives an impression that PDE9A is a solely decisive factor in ECC remodeling. Is this true? What about other pathways (CaMKII for example)?

Response 4

 Thank you again, upon re-reading this, we clearly see how this reads as a definitive statement that PDE9A is the only enzyme for ECC remodeling rather than another pathway to consider. This was not our intention as one of our references, Huang (13), explicitly states the function of calmodulin-dependent kinase II and its inhibition in RHF.

We have addressed this on Line 413, Page 9 Paragraph 8 by changing “Crucially, many of these molecular disturbances are downstream of impaired PKG signaling: when PDE9A is upregulated and cGMP is degraded, PKG activity falls, removing critical regulatory control over PLB, RyR2, and broader calcium cycling.” To read as “The elevated diastolic Ca²⁺ also promotes arrhythmias, a significant clinical complication of RHF. Many of these molecular disturbances, particularly involving PLB, RyR2, and calcium cycling, are plausibly downstream of impaired PKG signaling, a mechanism shown to be regulated by PDE9A in LV models. However, ECC remodeling is multifactorial, and pathways such as CaMKII activation and oxidative stress have known roles in RV dysfunction,” (13).

Comment 5

PKG phosphorylation ‘stabilizes RYR’. Is phosphorylation making RYR structure more stable or changes RYR sensitivity to Ca and thus reduce RYR Ca leak?

Response 5

 We appreciate you bringing this to our attention. ‘Stabilize’ is very vague and doesn’t distinguish between maintaining physical structure or modulation of function very well without additional context.

On line 303 of page 7 on our figure 1 description, we addressed this by changing, “PKG enhances calcium reuptake by increasing SERCA activity through phosphorylating phospholamban (PLB) and stabilizes ryanodine receptor 2 (RyR2)” to read as, “PKG enhances calcium reuptake by phosphorylating phospholamban (PLB) and reduces RyR2-mediated calcium leak, likely by modulating its gating behavior.

Comment 6

Figure 1. Arrows by PDE9A and pGC point in the opposite direction, i.e. indicated that PDE9 synthesize cGMP.

Response 6

We are very thankful that you caught this mistake. We have fixed this so that the image depicts proper function of PDE9A as a degrader of cGMP.

We greatly appreciate the time you have taken to read and review our paper. Your critique has aided us in presenting our stance as clearly and accurately as possible.

Round 3

Reviewer 1 Report

Comments and Suggestions for Authors

Thank you for answering my questions and concerns.